# Financial Fraud Detection with Entropy Computing

Babak Emami[1], Wesley Dyk[1], David Haycraft[1], Carrie Spear[1], Lac Nguyen[1], and Nicholas Chancellor[1]

Quantum Computing Inc (QCi), 5 Marine View Plaza Hoboken, NJ 07030 USA
bemami@quantumcomputinginc.com
http://www.quantumcomputinginc.com

**Abstract.** We introduce CVQBoost, a novel classification algorithm that leverages early hardware implementing Quantum Computing Inc's Entropy Quantum Computing (EQC) paradigm, Dirac-3. We apply CVQBoost to a fraud detection test case and benchmark its performance against XGBoost, a widely utilized ML method. Running on Dirac-3, CVQBoost demonstrates a significant runtime advantage over XGBoost, which we evaluate on high-performance hardware comprising up to 48 CPUs and four NVIDIA L4 GPUs using the RAPIDS AI framework. Our results show that CVQBoost maintains competitive accuracy (measured by AUC) while significantly reducing training time, particularly as dataset size and feature complexity increase. To assess scalability, we extend our study to large synthetic datasets ranging from 1M to 70M samples, demonstrating that CVQBoost on Dirac-3 is well-suited for large-scale classification tasks. These findings position CVQBoost as a promising alternative to gradient boosting methods, offering superior scalability and efficiency for high-dimensional ML applications such as fraud detection.

## 1 Introduction

Boosting is a powerful machine learning technique built on combining a set of "weak" classifiers which would be of little use individually to create a much stronger classifier which can perform well [5]. Boosting has been applied successfully to a variety of tasks, for example predicting chronic kidney disease [7], object detection [18], and email filtering [2] The fundamental task in boosting is deciding the weight to assign each weak classifier. A score is then determined form a linear weighted combination, and the score is used to determine the classification. Determining the weighting can be a hard task, since deciding which weak classifiers to include and how strongly to weight them should be done in such a way that it minimizes redundant information, leading to a naturally quadratic structure.

One advantage of boosting and in particular methods like QBoost and CVQBoost is that examining the highly weighted classifiers can allow researchers to understand how the algorithm is working. This is in contrast to methods such

as deep neural networks where such insight is not as readily accessible[1][20, 17]. A concrete example of finding explanations from boosting can be found in [18], where the authors examined a few of the highest weighted classifiers and found that they could be intuitively understood. In this example for face detection one of the most highly weighted weak classifiers was the presence of two dark rectangles surrounding a bright rectangle, corresponding to two eyes on either side of the bridge of someone's nose.

More explainable AI techniques such as ours (which broadly falls on the axis of "simplifying algorithms" in the strategies discussed in [20]) are likely to be particularly important in highly regulated sectors such as finance, where compliance is highly important. A concrete example here is European Union regulations which form part of the General Data Protection Regulation (GDPR) and give consumers a "right to explanation" about algorithmic decisions made related to them [9], in situations where this legislation is relevant, AI which gives a decision but no explanation of why the decision was made, could cause legal difficulties. Likewise, in scientific settings [17] (for example chemistry and computational biology), the goal is often to uncover natural laws rather than just make a single predictions. In these settings explainable AI is much more valuable than a "black box" which simply returns an answer, even if the answer it returns is reliably correct.

The QBoost algorithm [14], converts the problem of weighting the weak classifiers into a quadratic unconstrained binary optimization (QUBO) problem, where diagonal elements correspond to the quality of classification, and off-diagonals correspond to measures of correlation between the classifiers. An advantage of QBoost is that it is constructed to naturally match the form of problems which could be solved by quantum annealers. In this work we further extend this concept to CVQBoost, an algorithm which can naturally be applied to the Dirac optical computing hardware developed by Quantum Computing Inc. [16] which encodes into continuous variables. We further demonstrate the performance of this algorithm on a fraud detection use case.

We find that when applied using QCi's commercially-available Dirac-3 devices (hereafter referred to as Dirac-3) CVQBoost to achieve superior speed and power efficiency compared to a state-of-the-art conventional approach, XGBoost [4] while maintaining comparable accuracy. We particularly find highly favorable scaling of the runtime versus both number of features and training data samples.

In traditional boosting algorithms like AdaBoost [6] the weights of weak classifiers are iteratively adjusted based on their performance, aiming to minimize the overall classification error. CVQBoost innovates by using a novel optical solver to solve this optimization problem. It encodes the boosting task as a

---

[1] There is work on ways to make DNNs explainable (see citations in text), and this is an area of active research, but it would involve significant modifications to the methods. These explanations are furthermore often "fragile" in the sense that a small change to the data which shouldn't change anything, can completely change the explanation [8].

quadratic optimization problem, leveraging the Dirac-3's capability to effectively search rough optimization landscapes.

## 2    Implementation

### 2.1    Dirac-3 optimization Machine

Entropy quantum computing (EQC) has emerged as a promising optimization technique, using artificial dissipation to find the optimal solutions of a Hamiltonian [16]. In a photonic implementation, qudits are represented as superpositions of photon number states within time bins, which evolve in an optical fiber loop. This loop incorporates a target Hamiltonian as a dissipative operator, effectively simulating imaginary time evolution. During this process, high-energy eigenstates experience dissipation and decoherence, while lower-energy eigenstates are favored in the evolution. The latest iteration of EQC hardware at Quantum Computing Inc (QCi), the Dirac-3 system, is a hybrid architecture that combines the high-speed parallel processing capabilities of photonic systems with the precise control and programmability of electronic circuits. By exploiting the natural ability of photonics to manipulate complex optical fields for entropy-driven state evolution, alongside the use of electronics for state initialization, feedback, and fine-tuning, hybrid entropy computing enables efficient optimization. With analog computing approach, we explore if Dirac-3 provides advantages in accelerating training time while maintain low power consumption.

The relevant computational mode for this work is when it is used as a continuous quadratic solver, where it solves a problem of the form

$$\min_{w} \sum_i \sum_j J_{ij} w_i w_j + \sum_i C_i w_i \tag{1}$$

where $J$ and $C$ are a user specified problem definition and $w_i \in \mathbb{R}^+$ are continuous variables subject to $w_i > 0$ and an overall sum constraint

$$\sum_{i=1}^{N} w_i = 1. \tag{2}$$

Dirac-3 has a number of other capabilities such as higher order coupling, which are not used in this study.

### 2.2    Formulation

Our methods build on the conventional formulation of boosting [5], most similar to the quadratic formulation used in QBoost[14]. Let us assume that we have a collection of $N$ "weak" classifiers $h_i$ where $i = 1, 2, ..., N$. Depending on the method used to build weak classifiers, the values of $h_i$ can be discrete or continuous. The goal is to construct a "strong" classifier as a linear superposition of these weak classifiers, that is,

$$y = \sum_{i=1}^{N} w_i h_i(\mathbf{x}) \tag{3}$$

where $\mathbf{x}$ is a vector of input features and $y \in \{-1, 1\}$. The goal is to find $w_i$, continuous positive weights associated with the weak classifiers.

We use a training set $\{(\mathbf{x_s}, y_s) | s = 1, 2, ..., S\}$ of size $S$. We can determine optimal weights $w_i \geq 0$ by minimizing,

$$\min_{\mathbf{w}} \sum_{s=1}^{S} \left| \sum_{i=1}^{N} w_i h_i(\mathbf{x_s}) - y_s \right|^2 + \lambda \sum_{i=1}^{N} (w_i)^2 \tag{4}$$

where the regularization term $\lambda \sum_{i=1}^{N} (w_i)^2$ penalizes non-zero weights; $\lambda$ is the regularization coefficient which is used to control the relative importance of regularisation. Note that this is similar to the formulation in the QBoost [14] algorithm. The key difference is that we do not need to encode the weights into binary variables since the Dirac machine we use is based on a native continuum encoding. We now observe that the objective function in equation 4 can be cast (up to an irrelevant constant offset) in the form of equation 1 required by Dirac-3 by setting

$$J_{ij} = \sum_{s=1}^{S} h_i(\mathbf{x_s}) h_j(\mathbf{x_s}) + \delta_{ij} \lambda \tag{5}$$

$$C_i = -2 \sum_{s=1}^{S} y_s h_i(\mathbf{x_s}) \tag{6}$$

where $\delta_{ij}$ is a Kronecker delta, defined such that $\delta_{ii} = 1$ and $\delta_{i \neq j} = 0$.

### 2.3   Choices of Weak Classifiers

There are many ways to design a subset of weak classifiers. We have tested CVQBoost using logistic regression, decision tree, naive Bayesian, and Gaussian process classifiers. Each weak classifier is constructed using one or two of the features chosen from all features. This yields a set of weak classifiers that can be used to construct a strong classifier. In the present study, we have used logistic regression weak classifiers. In principle the highly weighted features within the most highly weighted regression classifiers could be used to explain the automated decisions made by our method, but we have elected not to examine this further as the main focus of the present work is to judge performance, not verify explainability.

### 2.4   Dataset

The dataset used in this study is the Kaggle Credit Card Fraud Detection dataset [13], a widely recognized resource for machine learning research, particularly in imbalanced datasets where the minority class constitutes a small fraction of the data.

This dataset comprises transactions made by European credit cardholders over a two-day period in September 2013. It contains approximately $200,000$ transactions, of which approximately $0.1\%$ are labeled as fraudulent. The significant class imbalance makes it ideal for exploring techniques tailored to imbalanced classification problems.

A total of 38 features are included in the dataset.

## 3   Results

In this section we present two sets of results. First in section 3.1 we analyze the accuracy of CVQBoost and find that at least when balancing has been applied the performance can be comparable to XGBoost. This is followed by analysis of the scaling of CVQBoost compared with XGBoost. We start with a simpler comparison with single core instances of XGBoost in section 3.2. Here we see substantial improvement, we also discuss the relative time CVQBoost spends on pre- and post- processing versus Dirac-3 runtime. We find that the time spent in the former scales modestly with both training data count and number of features, while the Dirac-3 time remains almost constant. Next in section 3.3, we present some scaling results compared with GPU and multicore processors on the Kaggle data. Finally, in section 3.4 we test scalability against training data count with a synthetic data set and compare with XGBoost implementations on muticore devices and GPUs. In this analysis we find favorable performance even when compared to state-of-the-art multi-core and GPU implementations of XGBoost. Unless stated otherwise, data are averaged over 10 runs and error bars are two standard deviations, so $95\%$ of data should lie within the bars assuming normally distributed results. The $95\%$ confidence intervals are therefore $1/\sqrt{10} \approx 1/3$ the size of the depicted bars.

### 3.1   Accuracy

Handling class imbalance is a central challenge in machine learning, particularly in applications such as fraud detection, where the minority class represents rare events. In the original dataset used for this study, fraud cases (the minority class) constitute less than $0.1\%$ of the data. To address this extreme imbalance, four widely-used data balancing strategies-ADASYN [10], SMOTE [3], SMOTE-SVM [15], and majority class downsampling-were applied to the training data [12]. The test data remained unchanged to ensure an unbiased evaluation of model performance. The accuracy of CVQBoost and XGBoost was assessed across six levels of class imbalance, characterized by minority-to-majority class ratios of

0.01, 0.02, 0.05, 0.1, 0.5, and 1.0. The Area Under the Curve (AUC)[1], calculated on the test data, was used to measure model accuracy. Figure 1 illustrates the trends for each balancing strategy.

The results reveal interesting patterns that provide insight into the behavior of the two algorithms under varying levels of class imbalance. When the balancing is minimal-that is, when the minority-to-majority class ratio remains small (0.01 or 0.02)-XGBoost achieves higher AUC scores compared to CVQBoost. This suggests that XGBoost is better able to leverage the heavily skewed class distribution in the early stages of balancing. However, as the class ratio increases and the training data becomes more balanced, CVQBoost's performance improves steadily and catches up to that of XGBoost, eventually achieving comparable AUC scores.

The effect of balancing strategies is particularly noticeable when using ADASYN. ADASYN generates synthetic samples by focusing on regions of the feature space that are prone to misclassification, thereby improving the representation of the minority class [10]. As the minority-to-majority class ratio increases, CVQBoost not only matches but eventually surpasses XGBoost in terms of AUC. This can be seen clearly in Figure 1(a), where CVQBoost achieves a slight but consistent improvement over XGBoost at higher class ratios. This suggests that CVQBoost is particularly effective at learning from the diversity introduced by ADASYN-generated synthetic samples.

A similar trend is observed with SMOTE and SMOTE-SVM. While XGBoost maintains a slight edge at smaller ratios, CVQBoost consistently narrows the gap as the balancing becomes more pronounced. The superior performance of CVQBoost at higher class ratios reflects its ability to generalize better when sufficient representation of the minority class is provided.

In the case of majority class downsampling, where no synthetic data is introduced and the majority class is simply reduced in size, CVQBoost algorithms demonstrate gradual improvement in AUC as the class ratio increases. However, the rate of improvement is less pronounced compared to the oversampling methods.

In summary CVQBoost becomes increasingly advantageous as class imbalance is mitigated, offering competitive performance for moderately to highly balanced training data. While the accuracy is comparable, in the findings we report later in this section we show that CVQBoost offers a major advantage in runtime even when compared to XGBoost run on state-of-the-art hardware.

### 3.2   Runtime versus single core XGBoost

**Impact of Training Data Count on Runtimes** Figure 2 presents the runtimes of CVQBoost and XGBoost across varying training dataset sizes, ranging from 1,000 to 150,000 samples. All runtimes are reported in seconds, with each experiment repeated at least 5 times and averaged to ensure reliability. The table provides both the mean and standard deviation of the runtimes, as well as a breakdown of CVQBoost's runtime components. Notably, the Dirac-3 component accounts for a substantial portion of the total runtime for CVQBoost.

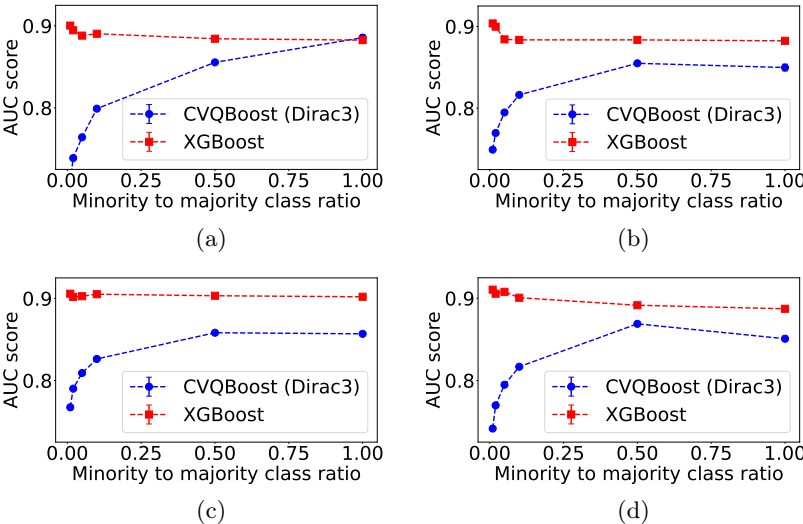

Fig. 1: AUC score of CVQBoost vs. minority-to-majority class ratio in training datasets using different balancing strategies: (a) ADASYN, (b) SMOTE, (c) SMOTE-SVM, and (d) majority class downsampling. The bars (obscured by the symbol at most points) are intervals in which 95% of instances should lie. These data can be found in tabular form in table 1 of the appendix.

A comparison between CVQBoost and XGBoost reveals that XGBoost achieves shorter runtimes on smaller datasets. However, as the training dataset size increases, XGBoost's runtime grows significantly faster than CVQBoost's. This trend is further illustrated in Figure 2a, that plots training runtimes against dataset size. Figure 2b shows a detailed barplot which visualizes the fraction of time spent by Dirac-3.

These results suggest that while XGBoost excels in speed for smaller datasets, CVQBoost offers a clear performance advantage as dataset sizes increase. Specifically, CVQBoost begins to outperform XGBoost when the training sample size approaches $20,000$ samples, highlighting its scalability for larger datasets.

**Number of Features** Figures 2c and 2d visualize the runtimes of CVQBoost and XGBoost for varying numbers of features, ranging from 5 to 38. All runtimes are reported in seconds, with each experiment repeated multiple times to ensure reliability.

Figure 2c illustrates the relationship between runtime and feature count for both methods. As the number of features increases, the runtimes of both CVQBoost and XGBoost grow. However, CVQBoost consistently demonstrates significantly shorter runtimes compared to XGBoost. Figure 2d breaks down the time spent by Dirac-3 versus other processes in CVQBoost. It is again notable that

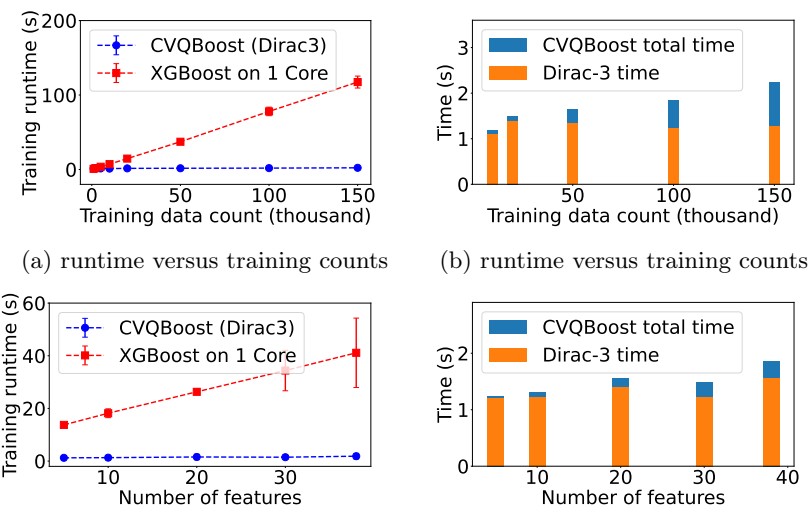

(a) runtime versus training counts     (b) runtime versus training counts

(c) runtime versus number of features  (d) runtime versus number of features

Fig. 2: (a) training runtime of CVQBoost and XGBoost vs. count of training data samples. The bars are intervals in which 95% of instances should lie. (b) fraction of CVQBoost runtime which was comprised by running on Dirac-3.(c) Training runtime of CVQBoost and XGBoost vs. number of features. The bars are intervals in which 95% of instances should lie. (d) fraction of CVQBoost runtime which was comprised by running on Dirac-3. These data can be found in tabular form in tables 2 (training counts) and 3 (feature counts) of the appendix.

the Dirac-3 component accounts for a substantial portion of the total runtime for CVQBoost.

### 3.3    Comparison with GPU and multicore implementations

**XGBoost on Multiple Cores** The performance of XGBoost was evaluated a high performance machine to assess its scalability in a multi-core environment. Figure 3 illustrates the training runtimes of XGBoost as a function of the number of cores utilized on a high-performance machine with 16 cores.

In Figure 3, diminishing returns are seen as more cores are added; while the runtime reduces with the addition of cores, the rate of improvement diminishes significantly beyond 8 cores, and further scaling provides only marginal benefits. There is also a degradation in runtime as the number of cores is increased beyond 8, this is likely due to inefficiencies caused by communication.

This saturation in scalability highlights a fundamental limitation of XGBoost in leveraging large numbers of cores for training speedup. In contrast, CVQBoost on the Dirac-3 platform exhibits remarkable scalability. Unlike XGBoost, which faces hardware limitations in multi-core environments, CVQBoost's design allows it to harness the power of optical hardware effectively, avoiding the saturation observed with multi-core scaling.

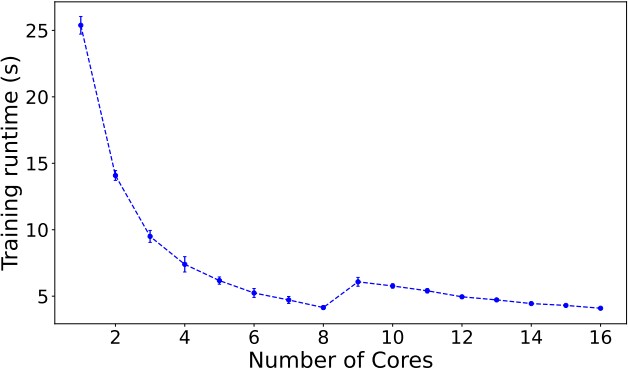

Fig. 3: Training runtime of XGBoost vs. number of cores on a high-performance machine with 16 cores. Intervals within which 95% of the data should lie are shown; training data count: $150,000$, number of features: 38.

As shown in Figure 3, using 8 cores for XGBoost appears to be optimal for minimizing training runtime.

Building on the scalability experiments, we conducted further tests to compare the performance of CVQBoost and XGBoost running in parallel on 8 cores, focusing on varying dataset sizes. As shown in Figure 4a, XGBoost achieves

shorter training runtimes for smaller datasets due to its efficient parallelization, however, as the data size increases CVQBoost runtimes become more favorable due to better scaling. It is also worth noting that CVQBoost runtimes are highly variable. Note that because each data point represents the average of 10 samples the 95% confidence intervals will be $1/\sqrt{10} \approx 1/3$ times the depicted bars. While the training runtimes of CVQBoost and parallelized XGBoost are similar for these relatively small datasets (ranging from $1,000$ to $150,000$ samples), experiments with much larger datasets reveal that CVQBoost scales significantly better with dataset size. This will be discussed further in Section 3.4, where a clearer scaling advantage can be seen at larger sizes.

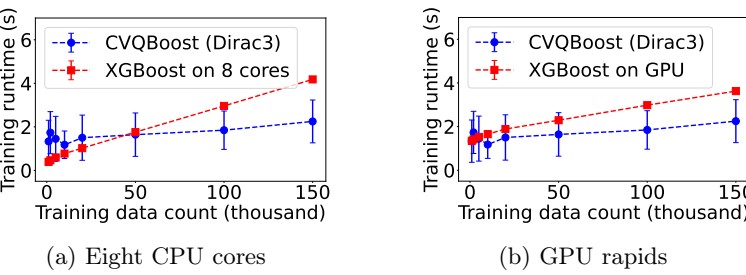

(a) Eight CPU cores                    (b) GPU rapids

Fig. 4: comparison of training runtimes for CVQBoost and XGBoost on (a) run on eight cores in parallel (b) GPU using the NVIDIA RAPIDS AI package. Bars represent intervals where 95% of data should lie. These data can be found in tabular form in tables (CPU) and 5 (GPU) of the appendix.

.

**XGBoost on GPU** To evaluate the performance of XGBoost in a GPU-accelerated environment, we ran the algorithm on an NVIDIA L4 GPU for various training dataset sizes, as shown in figure 4b. The NVIDIA RAPIDS AI package, version 24.12.00, was used. The GPU implementation demonstrates reduced runtimes compared to CVQBoost for smaller datasets. This efficiency arises from the GPU's ability to parallelize computations, which aligns well with the nature of XGBoost's gradient boosting framework. As the data size grows, CVQBoost shows slightly better scalability than XGBoost trained on GPU. While CVQBoost and XGBoost on GPU exhibit similar runtimes for the relatively small dataset sizes used here (ranging from $1,000$ to $150,000$ samples), further investigation reveals that CVQBoost scales significantly better on much larger datasets. This is discussed in Section 3.4.

### 3.4   Scalability on Synthetic Data

To evaluate the scalability of CVQBoost, we generated large-scale synthetic datasets using the *make_classification* functionality in the Scikit-Learn pack-

age. The dataset sizes range from 1M to 70M samples, with 80% allocated for training. Each dataset consists of 100 to 900 features.

Experiments were conducted on a high-performance computing system equipped with 48 CPU cores (2.64 GHz) and four NVIDIA L4 GPUs. Each GPU unit has 24GB of memory, totaling 96GB across all four units. XGBoost was tested under two configurations: (1) using 48 CPU cores and (2) running on four NVIDIA L4 GPUs via the NVIDIA RAPIDS AI framework. The results of these experiments can be seen in figure 5 with a comparison of scaling with training data count for 100 features in figure 5a and the analogous plot for 500 features in figure 5b. Both show a substantial advantage from CVQBoost. This is even true in the 500 feature case, where the XGBoost appears to be better able to take advantage of the massive parallelization provided by the GPU.

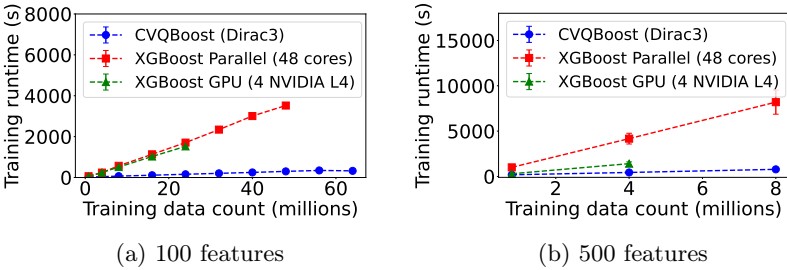

(a) 100 features                    (b) 500 features

Fig. 5: Training runtime of CVQBoost vs. XGBoost on 48 CPUs and four NVIDIA L4 GPUs. The bars are intervals in which 95% of instances should lie. CVQBoost exhibits superior scalability for large datasets.

Figure 6 further demonstrates that CVQBoost maintains linear scalability with increasing feature dimensions, whereas XGBoost exhibits quadratic runtime growth.

These results highlight the scalability advantage of CVQBoost for large-scale classification tasks, where computational efficiency is critical.

### 3.5   Training CVQBoost with Other Solvers

While the CVQBoost algorithm is designed to run on QCi's Dirac-3 device, the algorithm can be run using other solvers as well. We tested CVQBoost using two additional solvers: Hexaly [11], a commercial solver, and the Sequential Least Squares Programming (SLSQP) algorithm implemented in Scipy [19]. The objective of this comparison is to assess the scalability and efficiency of Dirac-3 relative to traditional optimization methods when solving Hamiltonians derived building a CVQBoost classification model.

The Hamiltonians used in this study were built from the synthetic datasets used in section 3.4 and range from 100 to 900 dimensions, with dynamic ranges

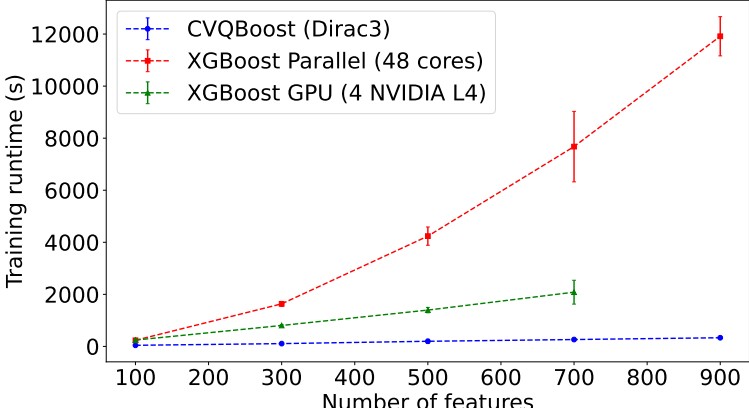

Fig. 6: Training runtime of CVQBoost and XGBoost vs. number of features. The bars are intervals in which 95% of instances should lie.

spanning 40 to 70 dB. The dynamic range, defined as the ratio between the largest and smallest coefficients in the Hamiltonian (expressed in decibels), significantly influences numerical stability and optimization difficulty. Notably, Dirac-3 has a known performance limitation at 23 dB, making it particularly relevant to evaluate how it performs when handling Hamiltonians that exceed this range.

For Hexaly, where we cannot directly control the optimality gap, we allow the solver to terminate based on either achieving the default optimality gap or reaching a predefined maximum iteration/runtime threshold. For a fair comparison, we use the default tolerances in Scipy's SLSQP solver and cap the maximum number of iterations for both solvers. This approach ensures that comparisons focus on fundamental scalability rather than fine-tuning solver-specific parameters.

Figure 7 presents a detailed comparison of Dirac-3 and the other solvers in terms of runtime and solution quality. The experiments were conducted across five trials per problem size, with the results averaged.

From Figure 7a, we observe that while the other solvers achieve shorter runtimes for small Hamiltonian sizes, their scalability deteriorates significantly for larger problem instances. They exhibit quadratic or worse runtime growth with the number of variables, whereas Dirac-3 maintains a more favorable linear scaling. At a Hamiltonian size of 900, Dirac-3 outperforms the other solvers in runtime efficiency.

Figure 7b compares the final solution energies, as measured against the quadratic objective functions. While all solvers achieve similar energy levels for small problem sizes (e.g., 100 variables), for larger problem instances, a slight energy gap emerges in favor of the other methods. Recall from our previous results that the solutions found by Dirac-3 are good enough quality to provide com-

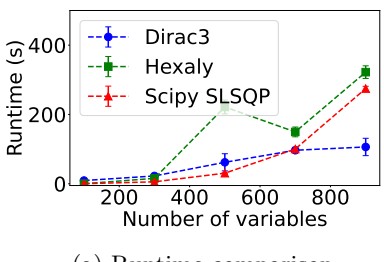
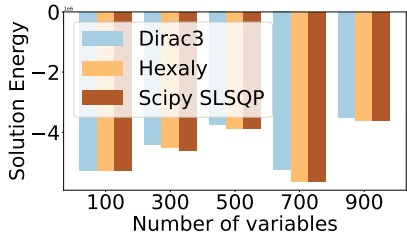

(a) Runtime comparison

(b) Solution energy comparison

Fig. 7: Comparison of Dirac-3, Hexaly, and Scipy (SLSQP) solvers in terms of (a) runtime scaling and (b) solution energy performance across different problem sizes.

petitive performance with more traditional boosting methods. This discrepancy is likely due to the large dynamic ranges of these Hamiltonians, which exceed the 23 dB operational limit of Dirac-3, potentially impacting its optimization performance.

## 4  Conclusions and outlook

In this work we have proposed and demonstrated an new boosting algorithm called CVQBoost which can leverage the unique features of QCi's Dirac-3 computer. The results show that comparable accuracy with traditional boosting methods (XGBoost) when balancing techniques have been applied. In particular, we found that CVQBoost can even exceed the performance of XGBoost when the ADASYN balancing strategy is used. This strategy involves generating artificial examples to improve regions of the feature space prone to misclassification suggesting that CVQBoost is particularly well adapted to learning from this method. We also find that CVQBoost can be trained much faster, even when compared to state-of-the-art implementations on GPU's and multicore systems.

While the specific demonstration used here was related to fraud detection, there will be many other potential applications. A particularly appealing future direction is in science, for example computational bioscience or chemistry. This is due to the potential for explainability of boosting techniques by examining the weak classifiers which were used in cases where the classifiers have more explanatory potential.

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

## A   Data in tabular form

| sampling strategy | minority to majority class ratio | cvqboost auc | xgboost auc |
|---|---|---|---|
| ADASYN | 0.01 | 0.7105 +/- 0.0 | 0.9003 +/- 0.0 |
| ADASYN | 0.02 | 0.7388 +/- 0.0 | 0.8946 +/- 0.0 |
| ADASYN | 0.05 | 0.7642 +/- 0.0 | 0.8881 +/- 0.0 |
| ADASYN | 0.1 | 0.799 +/- 0.0 | 0.8904 +/- 0.0 |
| ADASYN | 0.5 | 0.8555 +/- 0.0002 | 0.8843 +/- 0.0 |
| ADASYN | 1.0 | 0.8855 +/- 0.0019 | 0.8826 +/- 0.0 |
| SMOTE | 0.01 | 0.7493 +/- 0.0 | 0.9037 +/- 0.0 |
| SMOTE | 0.02 | 0.7697 +/- 0.0 | 0.8998 +/- 0.0 |
| SMOTE | 0.05 | 0.7948 +/- 0.0 | 0.8842 +/- 0.0 |
| SMOTE | 0.1 | 0.8164 +/- 0.0 | 0.8836 +/- 0.0 |
| SMOTE | 0.5 | 0.855 +/- 0.0004 | 0.8836 +/- 0.0 |
| SMOTE | 1.0 | 0.8498 +/- 0.0021 | 0.8824 +/- 0.0 |
| SVMSMOTE | 0.01 | 0.7674 +/- 0.0 | 0.9058 +/- 0.0 |
| SVMSMOTE | 0.02 | 0.7902 +/- 0.0 | 0.9019 +/- 0.0 |
| SVMSMOTE | 0.05 | 0.8092 +/- 0.0 | 0.9027 +/- 0.0 |
| SVMSMOTE | 0.1 | 0.8263 +/- 0.0 | 0.9051 +/- 0.0 |
| SVMSMOTE | 0.5 | 0.8582 +/- 0.0014 | 0.9033 +/- 0.0 |
| SVMSMOTE | 1.0 | 0.8569 +/- 0.001 | 0.902 +/- 0.0 |
| Downsampling | 0.01 | 0.7415 +/- 0.0 | 0.9105 +/- 0.0 |
| Downsampling | 0.02 | 0.7698 +/- 0.0 | 0.9056 +/- 0.0 |
| Downsampling | 0.05 | 0.7948 +/- 0.0 | 0.9079 +/- 0.0 |
| Downsampling | 0.1 | 0.8167 +/- 0.0 | 0.9007 +/- 0.0 |
| Downsampling | 0.5 | 0.8689 +/- 0.0004 | 0.8915 +/- 0.0 |
| Downsampling | 1.0 | 0.8508 +/- 0.0016 | 0.8872 +/- 0.0 |

Table 1: AUC scores for CVQBoost and XGBoost using different balancing strategies and minority to majority class ratios.

| train data count | cvqboost train time | xgboost train time | dirac3 time |
|---|---|---|---|
| 1000 | 1.3 +/- 0.5 | 0.9 +/- 0.1 | 1.3 +/- 0.5 |
| 2000 | 1.7 +/- 0.5 | 1.8 +/- 0.1 | 1.7 +/- 0.5 |
| 5000 | 1.4 +/- 0.5 | 3.7 +/- 0.1 | 1.4 +/- 0.5 |
| 10000 | 1.2 +/- 0.3 | 7.2 +/- 0.2 | 1.1 +/- 0.3 |
| 20000 | 1.8 +/- 0.9 | 14.6 +/- 0.3 | 1.7 +/- 0.9 |
| 50000 | 2.6 +/- 3.1 | 37.4 +/- 1.2 | 2.3 +/- 3.1 |
| 100000 | 1.8 +/- 0.4 | 78.0 +/- 2.9 | 1.2 +/- 0.4 |
| 150000 | 2.2 +/- 0.5 | 117.4 +/- 4.0 | 1.3 +/- 0.5 |

Table 2: Breakdown of CVQBoost and XGBoost runtimes for different training data counts; number of features: 38.

| num features | cvqboost train time | xgboost train time | dirac3 time |
|---|---|---|---|
| 5 | 1.2 +/- 0.4 | 13.7 +/- 0.3 | 1.2 +/- 0.4 |
| 10 | 1.3 +/- 0.4 | 18.2 +/- 0.8 | 1.2 +/- 0.4 |
| 20 | 1.6 +/- 0.5 | 26.3 +/- 0.5 | 1.4 +/- 0.5 |
| 30 | 1.5 +/- 0.4 | 34.4 +/- 3.8 | 1.2 +/- 0.4 |
| 38 | 1.9 +/- 0.5 | 41.1 +/- 6.6 | 1.6 +/- 0.5 |

Table 3: Breakdown of CVQBoost and XGBoost runtimes for different counts of features; training data count: 50,000.

| train data count | xgboost total train time |
|---|---|
| 1000 | 0.4 +/- 0.1 |
| 2000 | 0.5 +/- 0.0 |
| 5000 | 0.6 +/- 0.0 |
| 10000 | 0.8 +/- 0.0 |
| 20000 | 1.0 +/- 0.0 |
| 50000 | 1.8 +/- 0.0 |
| 100000 | 3.0 +/- 0.0 |
| 150000 | 4.2 +/- 0.1 |

Table 4: Parallelized XGBoost runtimes on 8 cores for different training data counts; number of features: 38.

| train data count | xgboost total train time |
|---|---|
| 1000 | 1.3 +/- 0.1 |
| 2000 | 1.4 +/- 0.0 |
| 5000 | 1.5 +/- 0.0 |
| 10000 | 1.7 +/- 0.0 |
| 20000 | 1.9 +/- 0.0 |
| 50000 | 2.3 +/- 0.0 |
| 100000 | 3.0 +/- 0.0 |
| 150000 | 3.6 +/- 0.0 |

Table 5: XGBoost runtimes on GPU for different training data counts; number of features: 38.

| num vars | convex | dynamic range | Dirac3 runtime | Hexaly runtime | Scipy runtime |
|---|---|---|---|---|---|
| 100 | False | 45.9 | 9.4 +/- 1.5 | 0.2 +/- 0.0 | 0.2 +/- 0.0 |
| 300 | False | 41.7 | 22.6 +/- 2.1 | 15.7 +/- 1.7 | 5.7 +/- 1.1 |
| 500 | False | 64.3 | 62.0 +/- 12.4 | 221.7 +/- 10.0 | 30.6 +/- 0.6 |
| 700 | False | 65.2 | 96.8 +/- 3.5 | 149.6 +/- 7.1 | 100.3 +/- 2.4 |
| 900 | False | 57.6 | 106.2 +/- 12.5 | 322.1 +/- 9.1 | 274.8 +/- 3.2 |

Table 6: Comparison of runtimes for Dirac 3, Hexaly, and Scipy (SLSQP) solvers across different Hamiltonian sizes.