# OpenReview forum: "Financial Fraud Detection with Entropy Computing"
_purdue.edu/Purdue_University/PQAI/2025/Symposium — PQAI 2025 Oral_

### Official Review · Reviewer_pfiv · 2025-07-24
**Review of "Financial Fraud Detection with Entropy Computing"**

**Rating:** 5
**Confidence:** 4

**Review:**

The manuscript "Financial Fraud Detection with Entropy Computing" is well-written and easy to understand, however there are some minor concerns:
1) There are few recent references (last 5 years) and few references related to quantum boosting methods.
2) It is not clear how the method compares to other QML approaches.
3) How is parameter optimization performed? Barren plateaus are typically a concern for QML models
4) Specifics on how to encode the financial fraud detection problem into a QUBO is needed to recreate this work- what do the coupling coefficients in the Ising model look like? Furthermore, penalizing constraints by adding them times a large constant $\lambda$ is known to be challenging, as the choice of $\lambda$ can artificially increase or decrease performance. What values were chosen here?
5) There is no code available on an open repository.

---

### Decision · Program_Chairs · 2025-07-29

Accept (Oral)